# Exploring the Papillomaviral Proteome to Identify Potential Candidates for a Chimeric Vaccine against Cervix Papilloma Using Immunomics and Computational Structural Vaccinology

**DOI:** 10.3390/v11010063

**Published:** 2019-01-15

**Authors:** Satyavani Kaliamurthi, Gurudeeban Selvaraj, Sathishkumar Chinnasamy, Qiankun Wang, Asma Sindhoo Nangraj, William CS Cho, Keren Gu, Dong-Qing Wei

**Affiliations:** 1Center of Interdisciplinary Science-Computational Life Sciences, College of Food Science and Engineering, Henan University of Technology, Zhengzhou 450001, China; satyavani.mkk@haut.edu.cn (S.K.); gurudeeb99@haut.edu.cn (G.S.); gkr@haut.edu.cn (K.G.); 2College of Chemistry, Chemical Engineering and Environment, Henan University of Technology, Zhengzhou 450001, China; 3The State Key Laboratory of Microbial Metabolism, College of Life Sciences and Biotechnology, Shanghai Jiao Tong University, Shanghai 200240, China; sathishimb@gmail.com or sathishimb@sjtu.edu.cn (S.C.); wangqiankun@sjtu.edu.cn (Q.W.); sindhoo_sind@yahoo.com (A.S.N.); 4Department of Clinical Oncology, Queen Elizabeth Hospital, Kowloon, Hong Kong; williamcscho@gmail.com or chocs@ha.org.hk

**Keywords:** cellular immunity, codon frequency distribution, HPV58, minor capsid protein, TLR agonist, prophylaxis

## Abstract

The human papillomavirus (HPV) 58 is considered to be the second most predominant genotype in cervical cancer incidents in China. HPV type-restriction, non-targeted delivery, and the highcost of existing vaccines necessitate continuing research on the HPV vaccine. We aimed to explore the papillomaviral proteome in order to identify potential candidates for a chimeric vaccine against cervix papilloma using computational immunology and structural vaccinology approaches. Two overlapped epitope segments (23–36) and (29–42) from the N-terminal region of the HPV58 minor capsid protein L2 are selected as capable of inducing both cellular and humoral immunity. In total, 318 amino acid lengths of the vaccine construct SGD58 contain adjuvants (Flagellin and RS09), two Th epitopes, and linkers. SGD58 is a stable protein that is soluble, antigenic, and non-allergenic. Homology modeling and the structural refinement of the best models of SGD58 and TLR5 found 96.8% and 93.9% favored regions in Rampage, respectively. The docking results demonstrated a HADDOCK score of −62.5 ± 7.6, the binding energy (−30 kcal/mol) and 44 interacting amino acid residues between SGD58-TLR5 complex. The docked complex are stable in 100 ns of simulation. The coding sequences of SGD58 also show elevated gene expression in *Escherichia coli* with 1.0 codon adaptation index and 59.92% glycine-cysteine content. We conclude that SGD58 may prompt the creation a vaccine against cervix papilloma.

## 1. Introduction

Currently, viral infection contributes to about 20% of the global burden of human cancer. Among other virus types, the human papillomavirus (HPV) is reported in about 5% of all human cancers, specifically infection associated with the cervix with 250,000 mortalities every year [1]. HPV with its double-stranded DNA contains a non-enveloped small virus, which infects a region of the cutaneous epithelial membrane (skin or integumentary system), or the mucous membrane (i.e., coated as an internal line in hollow spaced organs like the mouth, reproductive organ, urinary tract, or rectum) in the host system [2]. The nomenclature of HPV is distinguished by the International Committee on Taxonomy of Viruses (ICTV), and is based on the suggestion obtained from the study group of papillomavirus [3]. ICTV follows the practice of naming species after a specific virus, such as HPV16, while the related types, namely, the “type species,” are designated as strains within the species [3,4]. For example, the frequently used term “HPV species alpha-9” is a synonym for the ICTV term “HPV16 species”; it contains the HPV types 16, 31, 33, -35, 52, 58, and 67 strains, respectively. According to ICTV, the species HPV16 belongs to the family of Papillomaviridae and the genus of Alphapapillomavirus [3]. The genomic relationship between different cancer types has demonstrated that more than 99% of cervical cancer patients are infected with 15 different types of α-clade HPV, defined as “high-risk” or “oncogenic” genital HPVs. The α-clade HPV (6 and 11) causes genital warts while the remaining strains of HPV are related to the risk of cervical cancer. HPV infection is attributed to more than 50% of oropharyngeal and anogenital cancers [5]. Generally, the human immune system can clear the pathogenic infection caused by HPV within 2 years, but this also depends on the efficiency of an individual’s immune system and the invading type of HPV. However, in the case of a very weak immune system, it fails to remove the invading high-risk HPVs (hrHPV) that may lead to the development of cervical cancer [6,7]. hrHPV infections are responsible for causing more than 99% of precancerous cervical intraepithelial neoplasias (CIN) and invasive cervical cancers (ICC) [8,9,10]. In China, HPV-mediated cervical cancer is a substantial public health issue, with 1 million new cervical cancer incidences and 30,000 moralities registered every year [11,12]. In 2018, clinical, epidemiological, and clinicopathological studies reported HPV58 to be the second or third most predominant genotype in precancerous CIN I, II, III, and ICC lesions, a higher grade of squamous intraepithelial or cell carcinoma, and adenocarcinoma of HPV positive patients in different geographical regions of China [11]. Seven provinces of China have reported hrHPV-mediated cervical cancer incidences, namely, Guangdong, Liaocheng, Shanghai, Wenzhou, Wuhan, Southwestern China, and Western China [13,14,15,16,17,18,19]. Zhang et al. [20] reported that the HPV16 (6.4%) and HPV58 (5.3%) genotypes were predominantly found in males who had recently been involved in sex, in Shanghai.

Cervarix^®^, Gardasil^®^, and Gardasil 9^®^ are the three non-infectious prophylactic Food and Drug Administration (FDA)-approved HPV licensed subunit vaccines in active usage. These vaccines were developed from the major capsid L1 virus-like particles (VLPs) using recombinant DNA technology. Cervarix is a bivalent vaccine based on Baculovirus fermentation and it provides ~70% protection against HPV (16 and 18)-mediated cervical cancer but not against genital warts [21]. Gardasil is a quadrivalent HPV (6, 11, 16, and 18) vaccine based on yeast fermentation technology. It is efficiently used for the prevention of genital warts and gives ~70% protection for cervical cancer [22]. In 2009, the FDA approved a nine-valent Gardasil 9^®^ vaccine that provides protection to HPV types 6, 11, 16, 18, 31, 33, 45, 52, and 58. It has been used for both males and females in the age groups of 9–15 and 9–26 [23]. The new nine-valent vaccine exhibited a positive outcome in high-grade lesions in the absence of HPV (18 and 16) infections [24]. In October 2018, the FDA extended the use of Gardasil 9 to the age group of 27–45 among both the sexes. In addition, the L1 VLP (absence of viral genomic materials)-mediated vaccine production in the eukaryotic (ex. Baculovirus) host system is a complex and tedious process [25,26]. The main limitations of currently available prophylactic vaccines are as follows: they are strain specific, not therapeutic for patients already infected with HPV, they require multiple dosages, and are expensive [27,28]. In addition, the effective straightforward delivery of HPV vaccines can enhance the immunogenic potential against HPVs.

The implementation of the L2 minor capsid protein is a potential alternative in the HPV prophylactic vaccine production. Since the N-terminal region of the L2 protein is highly conserved in low-risk HPV (6 and 11) and 13 different hrHPVs, it is contrasted with the type-specific protection of L1 prophylactic VLPs [29]. The single copy of the L2 protein ~473 amino acids (AA) is present in each L1 capsomere, resulting in 72 copies per virion [30]. Incidentally, the L2 protein plays a vital role in L1 assembly into the VLPs and enhances the encapsidation of the double-stranded ~8kb circular viral genome [31]. Moreover, the full-length or polypeptides (1–8 or 11–200 AA in length) of the L2 protein enhance the production of neutralizing antibodies in vaccinated experimental models including mice, cattle, and rabbit [32,33,34]. To date, no L2 VLP-derived prophylactic vaccines have been approved in clinical trials [26] due to their limitation of weak immunogenicity, which imitates the incapability of multimerizing into VLPs.

With this information, in this study, we aimed to design the novel chimeric vaccine from the N-terminal region of the L2 sequence of the HPV58 targets to hrHPVs. The immunomic tools namely, the immune epitope database (IEDB) and NetMHCv4.0, Tepitool, CTLPred, PAComplex, IFNepitope, ABCPred, AllerTOP, AllergenFPv 1.0, ANTIGENpro, program of protein information resource (PIR), and epitopes conservancy were implemented to discover the overlapped epitope segment that is required to induce B-cell and T-cell immunity. Then, the chimeric vaccine (SGD58) was constructed using the overlapped epitope segments, Toll-like receptor (TLR) adjuvants, Th epitopes, and amino acid linkers. The physiochemical and immunological properties of the chimeric vaccine were validated using Protparam, SolPro, VaxiJen, and ANTIGENpro tools. In addition, homology modeling using iterative threading ASSEmbly refinement (I-TASSER), the Robetta beta full chain protein structure prediction server, structural refinement (GalaxyRefine and 3DRefine), and structural validation (protein structure analysis [ProSA], Ramachandran plot, and ERRAT) were performed to obtain the best three-dimensional (3D) model of the chimeric vaccine and the target TLR5 receptor. Then, the interaction of the chimeric vaccine with TLR5 and stability of this complex were determined through protein–protein (PP) docking and molecular dynamic (MD) simulation. Moreover, the virtual cloning and gene expression of the chimeric vaccine in *Escherichia coli (E. coli)* were analyzed to obtain a low-cost HPV vaccine.

## 2. Materials and Methods

### 2.1. Protein Sequences

The L2 protein of HPV58 (Accession No.: P26538), the Flagellin protein of *Salmonella enterica* serovar *Dublin* (Accession No.: Q06971), and human TLR5 (Accession No.: O60602) sequences were obtained from the Swiss-Prot reviewed universal protein knowledgebase (UniProt) database [35]. The designed chimeric vaccine was named SGD58, using the name of the first and principal authors along with the strain number.

### 2.2. Immunomics Analysis

#### 2.2.1. MHC-I Binding Epitope Segments Prediction

Two servers, IEDB and NetMHCv4.0, have been exploited for the identification of major histocompatibility complex class I (MHC-I) binding epitopes from the N-terminal region of the L2 sequence. Specific human MHC-I alleles such as the human leukocyte antigen (HLA)-A* (01:01, 02:01, 02:07, 11:01, 24:01), HLA-B* (46:01, 58:01) and HLA-C* (07:02, 12:03) were abundantly diagnosed in different regions of China, including Guizhou, Henan, Taihu River Basin, Tibetan, Yunnan, Wenzhou, and Wuhan. These alleles were selected for epitope prediction [36,37,38,39,40,41,42]. IEDB [43] is a freely available analysis resource with specified algorithms for the identification and determination of immunogenic epitopes. A consensus method was implemented to predict the MHC-I binding epitopes and its production pathway [44]. In this consensus method, three algorithms namely, the neural network (artificial), matrix method (stabilized), and peptide libraries (combinatorial) were combined to predict the promising CTL epitope segments. The epitopes involve proteasomal cleavage (pCle), a transporter associated with antigen processing (TAP), and the MHC-I binding pathway. The lowest percentile rank (<10%) indicated the good binding efficiency of epitopes with the restricted alleles. NetMHCV4.0 [45] is another potential tool implemented to find MHC-I binding peptides with the best Pearson’s correlation coefficient (PCC) of 0.895, based on the combined neural network. The strong and weak binding peptides were predicted based on the thresholds of <0.5 and <2, respectively.

#### 2.2.2. CTL Epitope and TCR -Peptide/Peptide -MHC Interfaces Prediction

The CTLPred tool is a direct method for the prediction of CTL epitope segments instead of MHC binders. The prominent combined approaches were implemented to find the epitopes, based on both the artificial neural networks (ANN) trained by Stuttgart neural network simulator (SNNS) and support vector machine (SVM) methods. The combined methods demonstrate a higher level of accuracy (75.8 %) compared with other individual methods of prediction such as ANN (72.2%) and SVM (75.4%). The default cutoff scores of 0.51 of ANN and 0.36 of SVM were used to find the epitopes or non-epitopes at which the sensitivity and specificity of the predictions are almost similar [46]. A web server PAComplex provides access to examine and visualize the TCR-peptide and peptide–MHC interface (pMHC), respectively. For a given viral protein query sequence, the joint Z-value is obtained with a threshold 2.5. Moreover, it allows the selection of only limited allotypes of MHC class I such as HLA-A*(02:01), HLA-B*(08:01, 35:01, 35:08, 44:05), and HLA-E, respectively. The Z-value was calculated using the following formula:*Jz* = *Z_MHC_* × *Z_TCR_*(1)
where *Z_MHC_* and *Z_TCR_* are the score of a TCR-pMHC complex, calculated by (E-*µ*)/*σ.* E denotes the interaction score, *µ* denotes the mean, and σ denotes the standard deviation from 10,000 random interfaces [47].

#### 2.2.3. MHC Class-II Binding Epitopes Prediction

MHC-II alleles include DQB1*(03:01, 03:03, 06:01), DRB1*(07, 09, 14:01:01, 15:01, 15:07:01), and DPB1*(05:01,05:02:01), specific to the Henan, Taihu River Basin, Tibetan, Yunnan, Wenzhou, and Wuhan provinces of China, which have been selected for epitope prediction [36,37,38,39,40,41,42]. The IEDB consensus approaches were used to predict MHC-II binding epitope segments using the neural network-based alignment, stabilized matrix methods-based alignment, and combinatorial library-based algorithms [48]. The peptides with the lowest percentile rank were considered to be of a higher binding affinity. Tepitool [49] is a tool from IEDB analysis resources, which provides accession to the prediction of both class I and II binders. The peptides which show the lowest percentile rank (IC_50_< or = 500nM) are potentially considered as higher affinity binding peptides.

#### 2.2.4. Interferon-Gamma (INF-γ) Inducing Epitope Prediction

IFNepitope (http://webs.iiitd.edu.in/raghava/ifnepitope/) that is a potential server useful for the prediction and design of INF-γ inducing epitopes. INF-γ inducing epitopes were identified based on motif-based SVM or hybrid algorithms. The hybrid method using residue or dipeptide composition shows 81.39% accuracy [50].

#### 2.2.5. Linear B-Cell Epitope Prediction

ABCPred is used to predict linear B-cell epitopes. It provides 65.93% of accuracy with the involvement of the recurrent neural network (RNN) algorithm. It consists of 700 B-cell and non-B-cell epitope segment datasets each with a length of 20 amino acids [51].

#### 2.2.6. Allergenicity Prediction

AllerTOP is the first proper alignment-free allergenicity server. In this, five machine learning methods such as partial least squares, logistic regression, decision tree, naive Bayes, or *k* nearest neighbors (*k*NN = 1) were implemented to find the allergen. It shows 88.7%, 90.7%, and 86.7% accuracy, specificity, and sensitivity, respectively [52]. AllergenFPv 1.0 is another essential tool for allergenicity prediction based on novel descriptor fingerprint approaches. Twenty naturally existing amino acids in the protein sequences were classified into five descriptors (E) such as E1 (hydrophobicity), E2 (size), E3 (helix-forming propensity), E4 (relative abundance of amino acids), and E5 (β-strand forming propensity). Based on this, the strings were transformed into normal vectors by auto cross covariance (ACC) transformation to find the allergen protein. It exhibits accuracy (87%), specificity (89%), and sensitivity (86%) [53].

#### 2.2.7. Antigenicity

ANTIGENpro is the potential alignment-free and sequence-based antigenicity prediction server with 79% accuracy and an area under curve (AUC) of 0.89. It shows results based on amino acid composition and the random-forest algorithm. The datasets were trained using 5-fold cross-validation. It consists of both protective antigen (193) and non-antigen (193) sequences. It predicts whether the given query epitope segments are antigenic or non-antigenic with their respective probability [54].

#### 2.2.8. Cross-Reactivity Analysis with Human Proteomes

The presence or absence of similarity in predicted epitope segments with the human proteome was analyzed using the peptide-matching program of PIR [55].

#### 2.2.9. Epitopes Conservancy Analysis

Epitopes conservancy (EC) and molecular evolutionary genetic analysis (MEGA) v7.0 tools were used to perform conservancy analysis. The EC tool [56] was employed to find the degree of conservancy of the epitope segments within the set of given hrHPV L2 protein sequences. The selected epitope segments of HPV58 with 15 hrHPV (16, 18, 31, 33, 35, 39, 45, 51, 52, 56, 59, 68, 69, 73, and 82) strains performed EC analysis. ClustalW-based multiple sequence alignment [57] was used to determine the sequence conservation of the overlapped epitope segments with 15 other hrHPV (16, 18, 31, 33, 35, 39, 45, 51, 52, 56, 59, 68, 69, 73, and 82) strains.

### 2.3. Chimeric Vaccine Design (SGD58) and Validation

#### 2.3.1. Assessment of the Physicochemical Properties of SGD58

The complete chimeric vaccine was designed by joining the optimized epitope segments (02), TLR adjuvants (02), and Th epitopes (02) with suitable amino acid linkers. Moreover, it is required to find the solubility of the designed chimeric vaccine on overexpression in *E. coli*. SOLpro is a useful tool to find the solubility of protein based on the two-stage SVM algorithm. It achieves an overall accuracy of 74%, which develops on standard evaluation metrics with 10-fold cross-validation. It predicts the query protein to be soluble or insoluble at *P* ≥ 0.5 [58]. A range of physiochemical characteristics of the designed chimeric vaccine was also determined through ProtParam [59].

#### 2.3.2. Determination of Antigenicity

VaxiJen is the primary server used for the prediction of antigenicity of the input sequence against different targets such as virus, bacteria, fungi, parasites, and tumors. Antigenicity was calculated based on the physicochemical properties of the protein sequences. Every target organism dataset contained 100 antigens and non-antigens. Moreover, the model organisms were validated using leave-one-out cross-validation (LOO-CV); it provides 89% accuracy and an AUC of 0.964 respectively, at the threshold of 0.4 [60].

#### 2.3.3. Analysis of the Tertiary Structure

##### 2.3.3.1. Homology Modeling

For homology modeling, I-TASSER and Robetta were the servers used to design the 3D structure of SGD58 and TLR5. I-TASSER is a potential server that depends on the secondary-structure-mediated program of “Profile-Profile threading alignment (PPA) and iterative implementation of the TASSER.” It has predicted a number of protein structures on request basis from 35 countries worldwide. For the query inputs, the user obtains the confidence score, TM score (topology similarity assessments of the two various protein structures), root-mean-square deviation (RMSD), and cluster density values. Nevertheless, the higher C-score (ranging from −5 to +2) determines the best model with a higher confidence level. Moreover, the 3D structure of the modeled protein was visualized using UCSF Chimera [61]. The Robetta beta server was used to predict the full chain protein structure. This server (http://robetta.bakerlab.org) gives automated tools for the analysis and prediction of the protein structure. Robetta provides both the *ab initio* and comparative models of protein domains. The comparative models are built from structures detected and aligned by HHSEARCH, SPARKS, and Raptor. The loop regions are assembled from fragments and optimized to fit the aligned template structures. The *de novo* models are built using the Robetta *de novo* protocol. For structure prediction, the submitted query sequences were analyzed minutely into putative domains. For domain prediction, a hierarchical screening method called “Ginzu” was used [62]. Besides, due to the unavailability of the crystal structure of TLR5, we have chosen TLR5 (PDB ID: 3J0A) as a template model to perform the homology modeling.

##### 2.3.3.2. 3D Modeled Structure Refinement

The high C-score model of the designed vaccine from the I-TASSER and model 3 from Robetta beta was further refined using the GalaxyRefine and 3DRefine tools. The GalaxyRefine is a tool that is accessible in the GalaxyWeb server: it is useful to refine the structure of a protein from the given query sequences based on template-based modeling, and undergoes loop and terminus portion refinement through the *ab initio* modeling method. The ninth critical assessment of techniques for protein structure prediction (CASP9) optimizes refinement and produces consistent core structures [63]. Another tool is 3Drefine, which prompts iteration analysis for ~300 amino acid residues efficiently in less than 5 min. It performs post-refinement model analysis with both or single MolProbity and random walk (RW) plus methods. The results are visualized using JSmol [64]. The top five models of each tool were used for further validation.

##### 2.3.3.3. 3D Refined Structures Validation

The refined 3D models from the above steps were validated using three interactive services namely, ProSA, Ramachandran plot analysis, and ERRAT. ProSA-web is a potential tool for the refinement, validation, prediction, and modeling of protein structures. It indicates the difference in the protein structures through the respective score and energy spot. It also facilitates the validation process of the protein structure that is acquired from X-ray scanning, nuclear magnetic resonance (NMR) spectroscopy analysis, and theoretical calculations. As an output, the *Z*-score corresponds to the overall feature of the validated model [65]. RAMPAGE tool is used to validate the percentage (%) of favored, allowed, and outlier region in the given query chimeric vaccine [66]. The statistics of noncovalent interactions between carbon, nitrogen, and oxygen atoms in the input sequence with best-resolution crystallographic structures were compared using the ERRAT tool. It implements an empirical atom-based approach for verification of the protein structure and is more sensitive to errors (1.5A) [67]. Similar steps were followed to validate the TLR5 model.

### 2.4. Conformational B-Cell Epitopes Prediction

DiscoTope 2.0 is a potential tool used to analyze the conformational (discontinuous) B-cell epitopes from the input sequence. It showed a highly significant prediction performance with an AUC of 0.824. The default −3.7 threshold limit provides significant specificity (0.75) and sensitivity (0.47). It was selected for the present analysis, and the final score was evaluated by the combined calculation of the propensity score (PS) and contact numbers [68].

### 2.5. Investigation of the Interaction between SGD58 and TLR5

#### 2.5.1. Protein–Protein (PP) Interaction of the SGD58 with TLR5

The PP interactions are the midpoint for all the biochemical pathways that are involved in the biological functions. HADDOCK v2.2 is a server used for the docking of PP complexes [69]. The scoring function was executed based on the weighted sum of the various energy terms (Van der Waals energy, electrostatic energy, desolvation energy, restraints energy and buried surface area). In addition, intermolecular contacts such as hydrogen bonds (HB) and those that are non-bonded were determined by using the program for automatically plotting protein–protein interactions (LIGPLOT-DIMPLOT) v 4.5.3 [70]. The validated best model of the vaccine construct (Robetta model 3) and TLR5 (Robetta model 5) was chosen as a ligand and receptor, respectively.

#### 2.5.2. MD Simulation

MD simulation determines the strength of the docked complex and the designed vaccine (SGD58). The GROMACS 5.1.2 package, with the CHARMM force field was used to perform MD simulation. The transferable intermolecular potential with three points (TIP3P) and simple point charge (SPC) in the cubic cell of the water model was resolved with protein, and with the addition of appropriate counter ions to satisfy electroneutrality. For the MD simulation, the system was implemented with volume-based canonical (NVT) and pressure-based isothermal-isobaric (NPT) ensembles [71]. The energy minimization of the system was performed with the steepest descent method, which facilitates 50,000 minimization steps and 1000 kJ mol^−1^ of tolerance. The Ewald method with a cutoff for short-range neighbor distance (1.0 nm), and Coulomb (1.0 nm) was used to calculate van der Waals (vdW) and electrostatic interactions [72]. SPC resolved the system and the final minimizations were calculated for a realistic structure concerning the geometry: and solvent orientations were used in the production of the MD simulation. SETTLE and LINCS algorithms were used to assist the geometry of the water molecules and bond angles [72,73]. The pressure of the system (300 K, 1bar) was embraced using the Parrinello–Rahman method and the temperature was regulated using the V-rescale method. Temperature and pressure equilibrated systems were employed for production run (100 ns) and time step (2 fs). The resulting structural coordinates were saved at every 2ps of an interval.

### 2.6. Analysis of Virtual Gene Expression and Cloning

EMBOSS Backtranseq v1.0 [74] is a suitable tool to uptake the query protein sequences, reverse translate, and return the optimizing coding sequences. Furthermore, the properties of the obtained coding sequences were analyzed to obtain the increased level of gene expression in the respective host. It is well known that the codon plays a crucial role in the expression of the recombinant proteins in various organisms (e.g., *E. coli*, *Homo sapiens*, *Saccharomyces cerevisiae*, *Mus musculus*, and *Rattus norvegicus*). The GenScript rare codon analysis [75] is a prominent tool for codon usage and its distribution analysis (codon adaptation index-CAI, glycine-cystine content-GC, and codon frequency distribution-CFD) in the individual expression host organism is based on the optimum Gene^TM^ algorithm. The endonuclease NdeI (N-terminal) and BamHI (C-terminal) restriction enzyme sites were added to the respective cloning sequences in the host (*E. coli*).

## 3. Results

### 3.1. Analysis of Selected Sequences

The retrieved L2 of HPV58, the Flagellin protein of *Salmonella enterica* serovar Dublin, and humanTLR5 contain 472, 505, and 858 amino acids residues, respectively. However, for the potential epitope prediction, an N-terminal region of 12-114 AA residues (highly conserved and virus surface exposed region) were selected from L2 protein sequences. In the case of the Flagellin protein, the N terminal (5–143 amino acids) and C-terminal (419–504 amino acids) regions were selected for the vaccine design.

### 3.2. Immunomics Analysis

#### 3.2.1. MHC-I Binding Epitopes

The region-specific MHC-I alleles in the Chinese population, namely, the HLA-A* (01:01, 02:01, 02:07, 11:01, 24:01), HLA-B* (46:01, 58:01), and HLA-C* (07:02, 12:03) restricted epitopes, were obtained using the IEDB consensus and NetMHC v.4.0 tools. The epitope segments from the N-terminal region of HPV 58 were overlapped using two different tools as shown in Appendix A.

#### 3.2.2. CTL Epitopes and TCR-Peptide/Peptide-MHC Interfaces

The overlapped epitope segments obtained from the above MHC-I prediction were compared with the results of both CTLPred [46] and PAComplex [47] servers. Furthermore, the shared epitope segments obtained from the CTLPred and PAComplex were used for epitope selection, and vaccine design as shown in Appendix A.

#### 3.2.3. MHC-II Binding and IFN-γ Producing Epitopes

The lowest percentile rank with strong binding affinity epitope segments with human MHC-II alleles, namely, the DQB1*(03:01, 03:03, 06:01), DRB1*(07, 09, 14:01:01, 15:01, 15:07:01), and DPB1*(05:01, 05:02:01) restricted epitopes were obtained using IEDB consensus and Tepitool servers. The overlapping promiscuous epitope segments from the above prediction (Appendix A) were selected and evaluated for their INF-γ production ability. The overlapped INF-γ producing CD4+ (MHC-II) epitope segments are as given in Appendix A. Therefore, the shared MHC-II epitope segments could produce IFN-γ against viral infection. Interestingly, the above-obtained overlapped CD4^+^ epitopes shared the CD8^+^ epitope segments.

#### 3.2.4. Continuous B-Cell Epitopes

ABCPred predicted the potential antigenic linear B-cell epitope segments. The overlapped B-cell epitopes and their respective position are given in Appendix A.

#### 3.2.5. Selection of the Overlapped Epitope Segments

According to the above results (MHC-1, CTL, MHC-11, INF-γ), only four potential antigenic epitope segments were selected from the HPV58 minor capsid protein. The epitope segments, namely,23–36, 30–43, 10–23, and 29–42 from the N-terminal region of the HPV58 L2 protein have been chosen for further studies (Table 1).

#### 3.2.6. Antigenicity, Allergenicity, and Cross-Reactivity of Selected Epitope Segments

The start and end positions, epitope segments, pro-inflammatory cytokines (INF-γ) productivity, allergenicity, antigenicity, and cross-reactivity with human proteomes of the selected overlapped HPV58 are given in Table 1. Among the four, only AllergenFP and AllerTOP declared segments (23–36 and 29–42) as non-allergen, respectively. AntigenPro shows the selected epitope segments having antigenic potential. In addition, the cross-reactivity result of the epitope segments with the human proteomes has a zero similarity level. It confirmed that there was no distinctive match of overlapped epitope segments as found in *Homo sapiens*. It indicated that these epitope segments would not cause or induce any autoimmune disease or disorders. Based on the above overall analysis, the 23–36 and 29–42 epitope segments were selected to design the vaccine construct.

#### 3.2.7. Epitopes Conservancy

In epitopes-based vaccine design, the conserved epitope segments would be essential in order to give wider cross-protection against various hrHPV strains. Appendix A gives the comprehensive analysis of overlapped EP (≥30%), positions, subsequences identity, and hrHPV. The conservation of selected epitopes has cross-protection to the 15 hrHPV as shown in Appendix A. The overlapped epitope segments KVEGTTIADQILRY_23-36_ and IADQILRYGSLGVF_29-42_ with 15-hrHPV strains are illustrated in Appendix A.

### 3.3. Vaccine Engineering

#### 3.3.1. Designing of Chimeric Vaccine SGD58

The complete vaccine construct consists of (1) two selected epitope sequences (23–36 and 29–42); and (2) two different TLR adjuvants, Flagellin and RS09. Flagellin is recognized as the TLR5 agonist that involves the activation of innate immunity. The head and tail of the vaccine construct contain the N-terminal (5-143 amino acids) and C-terminal regions (419-504 amino acids) of the Flagellin. In addition, RS09, a synthetic short peptide of the TLR4 ligand, is also used; (3) two different T helper (Th) epitopes, namelyPADRE and TpD, are used in the construct. The pan HLA-DR-binding epitope (PADRE) is frequently employed for synthetic or recombinant vaccine development and another universal epitope, TpD, which has 31 amino acids, is also used; and (4) the seven different parts in the vaccine construct were associated with the linkers GPGPG, AAY, and EAAAK. Finally, the designed chimeric vaccine (SGD58) with 318 amino acid sequences was drawn using illustrator for biological sequences (IBS) v1.0 as illustrated in Figure 1.

#### 3.3.2. Physiochemical and Immunological Properties of SGD58

The various physiochemical properties of the chimeric vaccine (molecular weight, theoretical pI, the total number of negatively and positively charged residues, extinction coefficient, estimated half-life, instability and aliphatic index, GRAVY, and solubility) are demonstrated in Table 2. The designed chimeric vaccine from the L2 protein of HPV58 was highly stable with the computed instability index of 35.88 (<40) and a molecular weight of 33.15 kDa. The highest amino acid composition in the chimeric vaccine is alanine (12.1%), serine (12.1%), leucine (9.3%), asparagine (8.4%), isoleucine (7.6%), glycine (7.0%), glutamine (6.5%), threonine (5.9%), aspartic acid (4.8%), valine (4.5%), glutamic acid (3.7%), proline (3.7%), arginine (3.4%), lysine (3.1%), phenylalanine (2.2%), tyrosine (2.2%), histidine (1.1%), methionine (1.4%), and cysteine (0.3%). The results of the SOLpro analysis indicate that the chimeric vaccine was soluble (0.62) on overexpression in *E. coli*. The antigenic score of the chimeric vaccine was demonstrated as 0.4301 (> threshold of 0.4) and 0.9438 using VaxiJen and ANTIGENpro (Table 2).

### 3.4. Structural Analysis

#### 3.4.1. Homology Modeling

I-TASSER homology modeling indicated that model 3 of the chimeric vaccine (SGD58) has the highest C-score of −0.39. In addition, the TM and RMSD scores of 0.66 and 7.2Å of model 3 represent the standard similarity and accuracy of the modeled structures. Notably, there is no 3D structure available for the TLR5 on the protein data bank. I-TASSER was used to model the 3D conformational structure of TLR5. The C-score, TM, and RMSD scores of the best model 1 of TLR5 are −0.35, 0.82 and 6.8Å, respectively. Therefore, model 1 is suggested as the best model with a higher confidence level. In addition, modeling with Robetta, we have obtained five different models for TLR5 and the chimeric vaccine candidate. Models 1 and 4 are *de novo* models while models 2, 3, and 5 are *ab initio* models. The Ginzu domain prediction confidence score for TLR5 is 0.9375, and for SGD58 is 0.6502. All the models were selected for structural refinement analysis.

#### 3.4.2. Structural Refinement

The acquired best-modeled 3D structures of SGD58 and TLR5 underwent structural refinement using servers GalaxyWEB and 3D refine. The GalaxyRefine program gives five best-refined models for the whole SGD58 and TLR5. In addition, the lowest 3Drefine score represents the good quality of the refined model, based on the 3D refine force field. All the refined models (1–5) of both GalaxyWEB and 3D refine were used for further structural validation.

#### 3.4.3. Structural Validation

The refined 3D structure obtained in the above section underwent quality improvement using three potential tools: ProSA-web, RAMPAGE, and the ERRAT. The Z-score (ProSA), overall quality factor (ERRAT), and the favored, allowed, and outlier region (RAMPAGE) of the validated 3D structure of SGD58 are given in Appendix A and TLR5 in Appendix A. Figure 2 illustrates the Z-score, overall quality factor and the favored, allowed, and outlier region of the selected best SGD58 model. From overall comparison of the obtained results, the Robetta model 3 of SGD58 (Appendix A) and the Robetta model 5 of TLR5 (Appendix A) using UCSF Chimera were selected for further analysis.

### 3.5. Conformational B-Cell Epitopes

The results of the Discotope 2.0 analysis demonstrated that 18 potential B-cell epitopes were obtained from 318 total residues. Appendix A explains the respective amino acid, residue with contact number, propensity, and Discotope score of the predicted B-cell epitopes.

### 3.6. Investigation of the Interaction between SGD58 and TLR5

#### 3.6.1. PP Docking Interaction

From the above findings, the best-refined model of SGD58 (Robetta model 3) and TLR5 (Robetta model 5) was used to perform molecular docking using the HADDOCK server. The binding cavity of TLR5 and Flagellin was obtained from a previous report [76,77]. The input TLR5 receptor contains 858 amino acids and SGD58 contains 2923 amino acids. The human TLR5 sequence contains 21 different leucine-rich repeats (LRR) segments with 443 amino acids. Flagellin contains two D1/D0 TLR-binding domains in the N and C terminals of the sequence.

The HADDOCK method directly permits the integration of biophysical information about the protein–protein complex in order to constrain docking. In this study, we docked the target receptor (TLR5) chimeric vaccine candidate (SGD58) to observe the interaction between the complexes. The HADDOCK method clustered 116 structures into 12 clusters, which represents 58.0% of the water-refined models that HADDOCK generated. The TLR5-SGD58 complex shows the highest HADDOCK score of −62.5 ± 7.6, representing the good affinity level between the target and vaccine. The buried surface area (BSA) of cluster 4 of the TLR5-SGD58 complex is 1914.4 ± 124.4, which indicates close proximity and a less water-exposed protein surface. The desolvation energy (43.1 ± 7.9), restraints violation energy (1192.9 ± 96.93), and BSA have a high-quality association with the docking score of the complex. The HADDOCK score, interaction energy, Van der Waals energy, electrostatic energy, desolvation energy, restraints violation energy, and BSA of the top tenclusters are given in Appendix A. In all, there are 44 hydrogen bonds between cluster 4 of the TLR5-SGD58 complex. The following amino acids, namely, Lys 148, Asp258, Ser 145, Arg 62, Asn 65, Gln 96, Arg 122, Asn 150, Asn 266, Tyr 120, Lys 177, Gly 76, Glu 80, Gln 72, Thr 73, Ser69, Asn 65, Gln 96, Arg 122, Thr 58, Asp 118, Gly 119, Asp 93, Asn 65, Ser 69, Arg 122, Asn 123, Asp 258, Thr 73, Gln 72, Glu 80, Asn 83, Lys 125, Gln 72, Ser 69, Asn 65, Arg 122, Ser 145, Gln 254, Gly 119, Glu 80, Asn 83, Thr 73, Arg 247, Glu 171, Asp 118, and His 143 act as interacting residues present in the best four clusters of the TLR5-SGD58 complex(Figure 3). Thus, the TLR5-SGD58 complex docking analysis and the intermolecular hydrogen bonding patterns confirm that the interaction of the chimeric vaccine candidate with the target TLR5 can induce both cellular and humoral immunity and inhibit HPV progression.

#### 3.6.2. MD Simulation

MD simulation demonstrated the stability of the TLR5_SGD58 docked complex in the active site of TLR5. RMSD is the known parameter by which to determine the obtained structure from the MD trajectory. This parameter was evaluated as a preliminary analysis of the backbone atoms of TLR5. The structure and dynamic proprieties of TLR5 were determined using the backbone RMSDs during the simulation period (Figure 4). The RMSD of TLR5 was gradually increased until 20 ns and a nearly 8–100 ns period with 3–4 nm of deviation, and after 30–80 ns, it was stable. The RMSD curve of SGD58 indicates an insignificant variation from 0–20 ns at 5.0 nm, and after 20 ns, it was stable. SGD58 has more a stable RMSD value compared to TLR5 (Figure 4A). The flexibility of each residue in the TLR5-SGD58 complex is calculated based on root mean square fluctuations (RMSF). Figure 4B shows that TLR5 has an insignificant variation of residues, which indicates that this molecule was stable during the MD simulation time of 100 ns. These residues have well-known interactions with the vaccine candidate. The ND1b domain (100–200 residues) of the Flagellin fragments of SGD58 shows a low flexibility, which can be attributed to their interaction with the TLR5 protein (Figure 4C). In addition, CD1 and CD0 domains have more fluctuations (200–250 residues). Figure 4D illustrates hydrogen bond interactions throughout the simulation period, to understand the binding efficiency of TLR5 with SGD58. The average number of hydrogen bond interactions was observed in 2.0 nm. Appendix A illustrates that the potential energy (PE), temperature, total energy (TE), and pressure of SGD58 was stable during the simulation period. The average TE of SGD58 is −7207307343.324 with a standard deviation of 4279.598082. In addition, the average PE of SGD58 is −8985342.697 with a standard deviation of 3370.264894. PE and TE attained equilibrium at a temperature of 300K. The result of the radius of gyration (Rg) analysis is shown in Appendix A. The simultaneous changes in the Rg plots of the complex with TLR5 (Appendix A) and SGD58 (Appendix A) indicate that the substantial nature of the complex frequently increases. The Rg plots compression of SGD58 with TLR5 is similar to the RMSD parameter, which indicates the effort of SGD58 to reach internal configuration in TLR5.

#### 3.6.3. Virtual Gene Expression and Cloning

For the given SGD58, the optimized reverse-translated coding sequences were obtained using the EMBOSS Backtranseq tool. The maximal protein expression of this optimized coding sequence in the host (*E. coli*) was analyzed by the GenScript’s OptimumGene^TM^ codon optimization tool. Appendix A illustrates the CAI, GC, and CFD of the gene transcript. The gene (reverse translated coding sequence of the vaccine construct) having an ideal CAI value of 1.00 (>0.8) is more suitable for the above expression (*E. coli*) in the host organism. Moreover, 59.92% of ideal GC content is presented in the gene (between 30% and 70%). However, values outside of these peak ranges would severely inhibit the transcriptional and translational efficiency of the gene products. The CFD value of the gene is 100%, representing their highest codon frequency distribution in the preferred expression organism.

## 4. Discussion

Immunomics is an integrative area of computer science and experimental immunology and plays a vital role in vaccine development. Immunomics tools and databases are used to forecast the target epitope segments to enhance CTL or B-cell immunity in a cost-effective manner and less experimental time [78,79,80,81,82]. The computational vaccine design involves the engineering of potential non-pathogenic epitope segments with adjuvants to enhance the function of the human immune system against dreadful diseases, including cervical cancer. Unlike conventional vaccines, peptide or epitope vaccines have several advantages; they are synthetic (pathogen-free), have less unwanted side effects, minimize accidental allergenic reactions, and design and predict peptides with self or non-self antigen to elicit and balance the immune responses [83,84]. HPV58 is considered as the most predominant genotype causing cervical cancer incidences in China. HPV has type-restriction, non-targeted delivery, and a high-cost of existing vaccines, which makescontinuing research on HPV vaccine development necessary. Therefore, this study aimed at the design of a chimeric vaccine *via* targeting HPV through immunomics, PP docking, and MD simulation approaches.

Earlier reports suggest that the L2 protein is majorly buried or hidden under the surface of native and matured virions [32,85,86]. The initial interactions between L1/L2 are hydrophobic with coverage of small stretches of amino acid sequences. It exhibits potential effects during *in vitro* assembly [87]. However, the structural relation of L2 minor capsid protein to L1 in the virion particles is not clearly known. In another study, Henio et al. [88] reported that the L1/L2 proteins of HPV have various antigenic epitope segments such as 32–81, 212–231, 272–291, and 347–381 amino acids, and these could be accessible on the surface of L1/L2 virus-like particles. In particular, the N-terminal region of the L2 protein is highly conserved and has diverse functions: it mainly participates in the attachment of the virus particle and its genome assembly in the host system. The N-terminal region 1–12 amino acids are in the DNA binding domain, the 9–12 amino acids are in the furin-cleavage site, and the 22–45 amino acids are in the cyclophilin-B and β-actin-binding domain [89,90,91]. The L2 protein can act as the prospective target to design the next-generation prophylactic vaccines for HPV [92]. This strategy is prominently supported by the early evidence, such as the production of cross-neutralizing antibodies RG1 [93], K4L2, and K18L2 [94] against the target sites (17–36 AA and 20–38 AA) of L2 in various experimental models. The weak immunogenicity of L2 is a significant obstacles in epitope-based vaccine development; to date, no L2-VLPs-based prophylactic vaccine has been approved for clinical application [30]. Therefore, in this study, we selected the L2 protein to predict the potential epitope candidate and enhance the immunogenicity using adjuvants to design a chimeric vaccine.

Adjuvants have different roles including up-regulation of the immune response, increased action of neutralizing antibodies, processing of cytosolic MHC class-I restricted peptides and target presentation to a specific receptor, acting as an immunogen, and application in the preparation of a single dose [95]. To increase the immunogenicity of L2-derived peptides, the selection of a suitable adjuvant plays a vital role in the vaccine design. Instead of mixing the appropriate adjuvant directly, designing the peptide vaccine candidate using suitable linkers and adjuvants could be highly effective. Alhydrogel^®^ adjuvant 2%, known as “alum,’’ is frequently used as an adjuvant in diphtheria-tetanus-pertussis (DTP), HPV, and hepatitis vaccination [96]. Although the alum adjuvants induced the Th2-mediated immune response, they are ineffective to the pathogen, which is indeed of the Th1 immunity. Later, the emulsion-based incomplete Freund’s adjuvant (IFA) was developed, which induces potential Th2, and little Th1-mediated immune responses [97]. However, the application of the emulsion-derived adjuvants is not supported well in the vaccination program due to the induction of autoimmune disorders and an unclear mode of action [98]. To overcome these issues TLR-based ligands were developed and achieved success in the generation of both Th1 and Th2 immune responses in the experimental models. Alphs et al. [99] achieved potential immunogenicity of the synthetic lipo-peptide (HPV16 L2) vaccine through fusion with the Th epitope and TLR ligand. Bacterial Flagellin is a potential TLR5 ligand, which can induce the production of both Th1 and Th2 immune responses.

It is frequently used as an adjuvant in the recombinant vaccine production when fusing with antigenic particles [100,101]. Flagellin, a TLR5 ligand, binds to the particular domain (Toll/interleukin-1 receptor) of the TLR5 receptor in humans. Notably, another newly developed and licensed adjuvant used for human vaccine development is the adjuvant system 04 (AS04). Moreover, AS04 was developed by a combination of 3-O-desacyl-4′-monophosphoryl lipid A (MPL), which is a prominent TLR4 agonist and aluminum salt. In the presence of Cervarix, the AS04 adjuvant induced the function of NF-kappa and cytokine synthesis in cancer cells and animal model. It leads to the appearance of increased antigen-loaded dendritic cells and monocytes followed by CD4^+^ and B-cells in the injection site [102]. Moreover, as a result of two-dose schedule trial in young girls (9–14 years), the HPV16/18 plus AS04 adjuvant vaccine (Cervarix) was highly immunogenic and has been approved clinically for the prevention of HPV infection, precancerous CIN (I/II/II), and cervical cancer [103]. In 18–25-year-old Chinese women, AS04 adjuvant vaccines were reported as having immunogenicity and an acceptable harmless profile from the randomized-controlled trial [104]. RS09 (short synthetic peptide segments “APPHALS”), is a ligand to TLR4. RS09 does not contain any toxicological effects, and is devoid of skin irritation, serious eye damage, and carcinogenic properties, etc. It successfully enhances the nuclear factor of kappa-light-chain-enhancer of activated B cell (NF-κB) translocation pathways and enhances the pro-inflammatory cytokine and antibody serum concentration in macrophage cells and animal models [105]. TLR adjuvants play an advanced role in commercial vaccines [106]. Two universal Th epitopes (PADRE and TpD) were added to the chimeric vaccine to resolve the deficiency of Th responses. The pan HLA DR-binding epitope is known as “PADRE”. It has a binding affinity with more than 15 MHC class-II allotypes and induces proliferative CD4^+^ in peripheral blood mononuclear cells (PBMC) from humans [107]. In this manner, it explains the issue raised by HLA polymorphism in the human population [108]. It is extensively studied for synthetic peptide-based vaccine development in C57BL/6 cervical cancer models and Phase I/II clinical trials [109,110,111]. TpD is another universal memory T-cell helper peptide. The immunization of TpD produces a promising antibody and enhances long-term CD4^+^ immune response in mice, *Rhesus macaques*, cynomolgus monkeys, and PBMC in humans [112]. Therefore, we have chosen two different Th epitopes (PADRE and TpD) and two TLR adjuvants (Flagellin and RS04) to enhance the immunogenicity in the chimeric vaccine.

A small flexible linker sequence was employed to join various segments of epitopes, the TLR agonist, and the Th epitopes in the vaccine construct. In this study, GGS linker was used to join the various segments in the vaccine design. The GGS linker facilitates the natural rotation or movement of the epitope segments and adjuvants and ameliorates their free identification by the surface receptor molecules [113,114]. GGS linkers contain nonpolar glycine (Gly) and polar serine (Ser) amino acid residues, which prohibit unnecessary complex formations between the linked partners and retain the function of the chimeric vaccine. A GGS spacer was presented in both natural and artificial linkers, to either increase the stability of the binding domain partners or stabilize the PP complex [115].

The targeted delivery of a vaccine can improve the efficiency and achieve a better outcome. In this study, weselected TLR5 as the target for the chimeric vaccine. Innate immunity-inducing TLR5 are mainly expressed on antigen presenting monocytes and dendritic cells while they encounter the entry of pathogenic microbes [116]. Moreover, TLRs are a well-categorized pattern recognition receptor (PPR) family, which involves the sensing of invading virulent pathogens entry into the host [117]. Horseshoe-shaped LRR are present in each TLRs conserved fold for binding to their respective legends [118]. Numerous studies report that after the binding of TLR5 to the specific ligand, it induces the myeloid differentiation gene 88 (MYD88), which triggers activation of the tumor downstream signaling pathways including NF-κB, mitogen-associated protein kinase (MAPK), and interferon regulatory factors (IRFs) [119]. Once the TLRs recognize their ligand, they become active and induces the production of pro-inflammatory cytokines such as tumor necrosis factor (TNF), interleukins (IL), and INFs [117,118]. In this manner, the host cell increases the capacity to eliminate the invading pathogens. Kim et al. [120] reported that TLR5 is a potential biomarker for the malignant transformation of cervical squamous cells. Therefore, in this study, we selected TLR5 as a target for the chimeric vaccine and its efficiency was determined using PP docking and MD simulation.

TLR5 is an excellent receptor for Flagellin, which is the major component of the bacterial Flagella [116]. Flagellin adjuvant has been extensively used in experimental HPV vaccination. Interestingly, when the host cell responds to Flagellin, TLR5 induces B-cell differentiation into the plasma producing B-cells. Earlier reports demonstrate the significance of Flagellin fused L2-multimer vaccines in experimental rabbits and mice [121,122]. Flagellin is madeup of four important domains: D0, D1, D2, and D3. The D0 and D1 domains are composed of highly conserved N-terminal (1 to 200 amino acids) and C-terminal (405–494 amino acids) regions, which is important for TLR5 agonist action. In addition, the centered hypervariable D2 and D3 regions show the vast differences, by their size and composition, among the various bacterial microorganisms [123,124,125]. Owing to the higher antigenicity and toxicity caused by the central D2/D3 domain, this antigenic part is removed or replaced by the optimized epitope segments or different adjuvants in the vaccine design. D2/D3 antigen replacement in Flagellin enhances mucosa-immunoglobulin productions in the experimental animal models through intranasal immunization [126]. Therefore, we selected the N and C-terminal regions of Flagellin in the design of the chimeric vaccine. Forstneric et al. [125] reported the appropriate identification of Flagellin by the homology modeled hTLR5 and mTLR5. The crystallographic complex structure of zebra fish TLR5 with the domains (D1, D2 or D3) of *Salmonella* sp. were also studied [77]. However, there was a lack of availability of the crystal structure of TLR5 until now. Therefore, in this study, homology modeling, structural refinement, and validation were performed and found using the Robetta model 5 of TLR5 obtained using the Robetta, 3DGalaxyRefine, ProSA-web, RAMPAGE, and ERRAT. Moreover, TLR5 is greatly presented in vertebrates [77,125,127]. It facilitates the PP interaction analysis of TLR5 with Flagellin using HADDOCK, as shown in Figure 4b. It shows the interacting residues of this complex observed between the LRR region of TLR5 and the D0/D1 domain of Flagellin. This finding is significantly supported by earlier reports [125,127] concerning TLR5 recognition of the D0 of Flagellin by the inflammasome receptor, for preventing the immune escape of invading pathogenic strains. The MD simulation results depict a constant and stable interaction between SGD58 and TLR5. During the MD simulation study, TLR5 was stable after 15 ns, whereas SGD58 exhibited insignificant variations and was then stable after 10 ns. The structural changes were observed to have gained the optimal sustainability of SGD58 and TLR5. In addition, very slight changes were noticed in the D0/D1 domain regions of SGD58. Finally, the results of the codon optimization and virtual cloning confirms the translated chimeric vaccine sequence in *E. coli* to be capable of regulating the higher level of gene expression. The successful expression of the designed virus-like particles of HPV in *E. coli* is reported in earlier studies [128,129], which can enhance the production of the vaccine at a cheaper cost.

From this report, the new chimeric vaccine candidate was engineered using various immunomics tools, PP docking, and MD simulation, which can reduce the experimental cost and time. The designed chimeric vaccine SGD58, has appropriate structurally refined, physiochemical, and immunological properties that can produce humoral and cellular immune responses against HPV. The designed chimeric vaccine has cross-production with the 15 different hrHPV strains. Further experimental investigation is planned to determine the efficiency of the chimeric vaccine, especially allele-specific for the Chinese population.

## Figures and Tables

**Figure 1 viruses-11-00063-f001:**
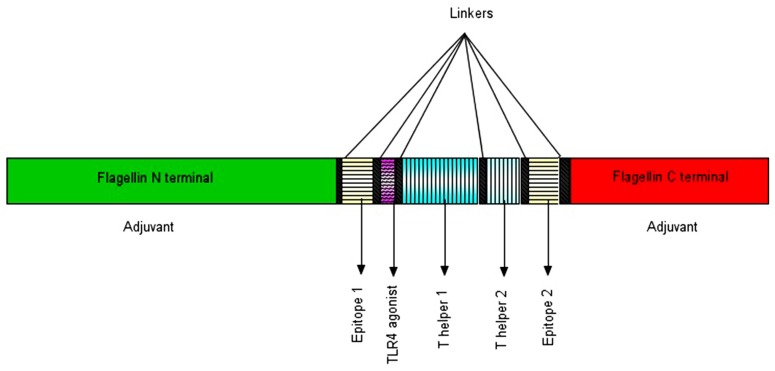
The designed vaccine construct 58 (SGD58). The SGD58 contains seven different segments, namelytwo different adjuvants (N and C-terminal region of Flagellin-TLR5 agonist); two epitope segments (23–36 and 29–42); adjuvant RS09 (TLR4 agonist) and two Th epitopes (PADRE and TpD). All the segments were joined together by using the following linkers (GPGPG, AAY or EAAAK).

**Figure 2 viruses-11-00063-f002:**
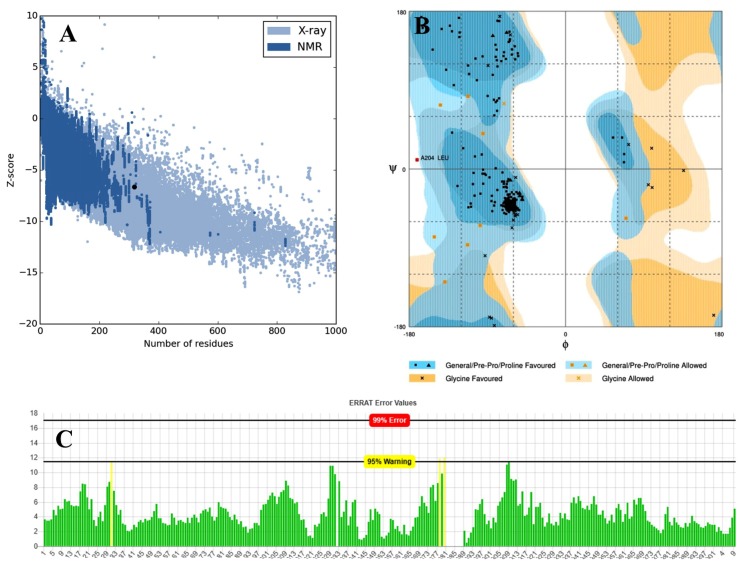
Validation outcome of the refined 3D structure of SGD58. (**A**)The ProSA Z-score of the refined model is -6.65, which is shownby the dark black color spot. The values are presented in the range of native protein conformation. The dark blue and light blue color region represents the Nuclear magnetic resonance and X-ray spectroscopy determination of the experimental protein chains in the protein database (PDB). The X- and Y-axis represent the number of amino acid residues and Z-scores respectively; (**B**) In the Ramachandran plot of the refined model, we illustrated the favored in green circle (96.8%), the allowed in triangle (2.8%) and the outlier in yellow shaded circle (0.3%) regions. (**C**) The ERRAT Plot shows that the overall high quality factor of the refined SGD58 is 99.0033. * On the error axis, two lines are drawn to indicate the confidence with which it is possible to reject regions that exceed that error value. ** Expressed as the percentage of the protein for which the calculated error value falls below the 95% rejection limit. Good high resolution structures generally produce values around 95% or higher. For lower resolutions (2.5 to 3A), the average overall quality factor is around 91%.

**Figure 3 viruses-11-00063-f003:**
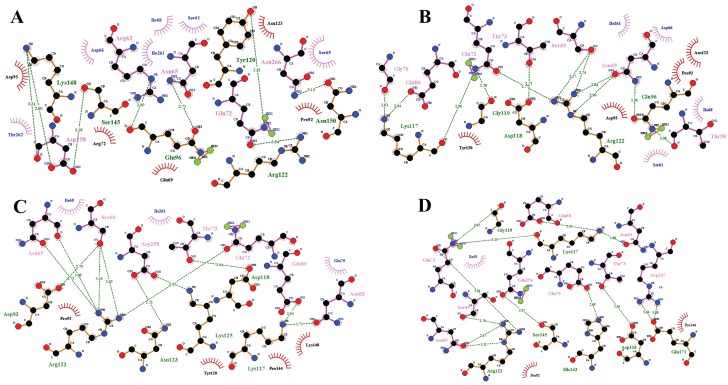
LIGPLOT prepared interacting residues in the TLR5_SGD58 complex. (**A**–**D**) represent the best structures of the TLR5_SGD58 cluster. The color-coding represents the TLR5 in brown color and the SGD58 in pink color. The dashed lines in green color denote hydrogen-bonding interactions.

**Figure 4 viruses-11-00063-f004:**
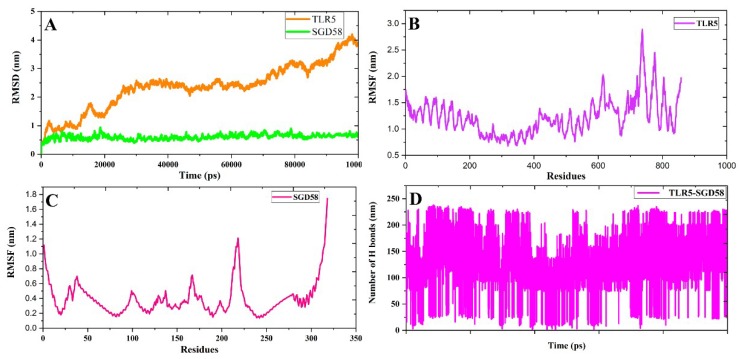
(**A**) The backbone root mean square deviation profile of TLR5 and SGD58 is depicted for the entire 100 ns. (**B**) The root mean square fluctuation of TLR5. (**C**) The root mean square fluctuation of SGD58. (**D**) The total number of intermolecular H bond interactions between SGD58 in complex with TLR5.

**Table 1 viruses-11-00063-t001:** Selection of overlapped epitope segments from the N-terminal region of HPV58.

Start	End	Overlapped Epitope Segments	IFN-γ Producing Epitopes (Potentiality/ Value)	AllergenFP	AllerTOP	AntigenPro	Cross-Reactivity with Human Proteomes
23	36	**KVEGTTIADQILRY**	+0.528	Non-allergen	Non-allergen	Antigen	Similarity level zero
30	43	ADQILRYGSLGVFF	+1.000	Non-allergen	-	Antigen	Similarity level zero
10	23	CKASGTCPPDVIPK	+1.000	-	-	Antigen	Similarity level zero
29	42	**IADQILRYGSLGVF**	+1.000	Non-allergen	Non-allergen	Antigen	Similarity level zero

The results of the analysis showed four overlapped epitope segments, among these the two epitope segments (23–36 and 29–42) indicated in bold were selected for the vaccine construction. The IFN-γ production ability of the overlapped epitope segments is presented in positive values.

**Table 2 viruses-11-00063-t002:** Evaluation of the various physiochemical and immunological properties of the chimeric vaccine SGD58.

Properties	Results/Values
Number of amino acids	318
Molecular weight	33,394.15
Theoretical pI	8.00
Total number of negatively charged residues (Asp + Glu)	24
Total number of positively charged residues (Arg + Lys)	25
Extinction coefficient M-1 cm-1	12,950
Half-life	20 h (Mammalian reticulocytes, *in vitro*), 30 min (yeast, *in vivo*) and >10h (*E.coli*, *in vivo*)
Instability index	35.88 (Indicates protein as stable)
Aliphatic index	94.62
Grand average of hydropathicity (GRAVY)	−0.190
Solubility (SolPro)	Soluble with probability 0.621085
Antigenicity (VaxiJen)	0.4301(Probable antigen)
Antigenicity (AntigenPro)	0.943820 (Probable antigen)

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
