# Peer review of "Exploring the Papillomaviral Proteome to Identify Potential Candidates for a Chimeric Vaccine against Cervix Papilloma Using Immunomics and Computational Structural Vaccinology"

_viruses, 2019, doi:10.3390/v11010063_

Reviewer 1 Report

Kaliamurthi et al. design, construct and evaluate a chimeric vaccine against a human papillomavirus implicated in the development of cervix papilloma, HPV58, using a combination of computational and experimental approaches. The newly engineered vaccine targets the human TLR5 receptor and, in contrast to existing L1-based vaccines, is based on epitopes in the papillomavirus L2 protein.

Computer-assisted vaccine design is of general importance and is expected to have great promises due to considerable cost and time reductions compared to traditional approaches. Overall, this is a well-conducted study, and I hope that the following suggestions will help to further improve the manuscript.

Please add some information about taxonomic classification of the analyzed papillomaviruses to the Introduction. It should for instance be acknowledged that HPV56 belongs to the genus Alphapapillomavirus defined by the ICTV. Also the ICTV species could be mentioned.

From my point of view, the manuscript would benefit from adding another illustration showing a multiple sequence alignment of the L2 protein region that encompasses the two epitope segments highlighted in bold in Table 1. I suggest to include all alphapapillomaviruses, or maybe even all human papillomaviruses if feasible, into this alignment. The alignment would show which sites in the epitope sequences are strongly or weakly conserved and may give an idea about whether or not the chimeric vaccine designed by the authors may be effective against other human papillomaviruses.

Line 613: „... an ideal chimeric vaccine was engineered ...“. I suggest to tone down this statement.          

Author Response

1)      Please add some information about taxonomic classification of the analyzed papillomaviruses to the Introduction. It should for instance be acknowledged that HPV56 belongs to the genus Alphapapillomavirus defined by the ICTV. Also the ICTV species could be mentioned.

The respective suggestions has been included in the text as follows

The nomenclature of HPV is distinguished by the International Committee on Taxonomy of Viruses (ICTV), and is based on the suggestion obtained from the study group of papillomavirus [3]. ICTV follows the practice of naming species after a specific virus, such as HPV16, while the related types, including the “type species,” are designated as strains within the species [4, 3]. For example, the frequently used term “HPV species alpha-9” is a synonym for the ICTV term “HPV16 species”; it contains the HPV types -16, -31, -33, -35, -52, -58, and -67 strains, respectively. According to ICTV, the species HPV16 belongs to the family of Papillomaviridae and the genus of Alphapapillomavirus [3].

2)      From my point of view, the manuscript would benefit from adding another illustration showing a multiple sequence alignment of the L2 protein region that encompasses the two epitope segments highlighted in bold in Table 1. I suggest to include all alphapapillomaviruses, or maybe even all human papillomaviruses if feasible, into this alignment. The alignment would show which sites in the epitope sequences are strongly or weakly conserved and may give an idea about whether or not the chimeric vaccine designed by the authors may be effective against other human papillomaviruses.

The epitope conservancy in the 15 hrHPV L2 protein has been analysed and showed in the figure as follows

Methods: Epitopes conservancy (EC) and molecular evolutionary genetic analysis (MEGA) v7.0 tools were used. ClustalW-based multiple sequence alignment [57] was used to determine the sequence conservation of the overlapped epitope segments with 15 other hrHPV (16, 18, 31, 33, 35, 39, 45, 51, 52, 56, 59, 68, 69, 73, and 82) strains.

Result: The overlapped epitope segments KVEGTTIADQILRY23-36 and IADQILRYGSLGVF29-42 with 15-hrHPV strains are illustrated in supplementary figure.

Figure S1b. The sequence conservation of overlapped epitope segments of HPV58 as KVEGTTIADQILRY 23-36 and IADQILRYGSLGVF 29-42 was done using MEGA v7.0. Clustal analysis was analyzed using USCF Chimera. The epitope segments are represented by black rectangular boxes.

3)      Line 613: „... an ideal chimeric vaccine was engineered ...“. I suggest to tone down this statement.

The statement has been changed as follows

From this report, new chimeric vaccine candidate was engineered using various immunomics tools, PP docking, and MD simulation, which can reduce the experimental cost and time.

Reviewer 2 Report

In this manuscript, Kaliamurthi and colleagues used computational immunology and structural computational vaccinology approaches, to design a theoretical vaccine against the Human papillomavirus type 58 (HPV58), which is relatively prevalent in China and other Asian countries. The authors identified two peptides from the capsid protein L2 and aimed at demonstrate that the respective peptides can be engineered to generate a vaccine candidate. The authors used a variety of software to identify the peptides of interest and claim that the identified peptides should generate a B cell response and be well presented by class I and class II MHC. Authors continued their investigation using I-Tasser in combination with some refinement software to generate a 3D model of the vaccine candidate. It is unfortunate that the authors did/could not validate their model with mice immunization. The overall study is interesting and address a very specific issue, even if the concept of epitope-focused immunogen was already described. Here are some conceptual and technical points that could/should be addressed to ameliorate the manuscript.

Major points:

1- The authors should edit the title of the manuscript, to highlight that the study is fully performed using in silico methodology.  For example, the authors did not perform structural vaccinology, but computational structural vaccinology.

2- It is interesting that the peptides predicted for MHC binding are present in the N-terminus of L2, knowing that the rest of L2 is predominantly hidden below the surface of native virions (Buck CB, J Virol 2008). The authors should discuss it in their manuscript.

3- I am not fully convinced by the structural modelling performed by the authors. Indeed, many outliers are visible in the Ramachandran plot and the model should reach a score >95% for all residues present in Ramachandran favored regions. The errata plot also indicates some general problem with structural modeling with several residue over the rejection limit of 95%.

It would also be useful for the authors to show the ERRATA on the full-length model and not only the first 140 amino acids.

Therefore, I invite the authors to perform a Homology modeling using another algorithm-based software. I would propose Modeler (B. Webb et al. Current Protocols in Bioinformatics 2016) or an ab initio prediction using Rosetta (Ovchinnikov S, Elife. 2015) or Robetta software (http://robetta.bakerlab.org/) for the two epitopes chosen to see how the model generated by these algorithms compare to the I-Tasser model.

Minor points:

1- There are some typos in the text that should be corrected.

2- Figure 2 could be improved (colors)

3- For Figure 3, it is very difficult to see anything. The authors should show the docking results from different angles. Front, top… and increase the resolution of Fig3b.

Author Response

1)      The authors should edit the title of the manuscript, to highlight that the study is fully performed using in silico methodology.  For example, the authors did not perform structural vaccinology, but computational structural vaccinology.

The title has been changed as follows

Exploring papillomaviral proteome to identify potential candidates for chimeric vaccine against cervix papilloma using immunomics and computational structural vaccinology

2)      It is interesting that the peptides predicted for MHC binding are present in the N-terminus of L2, knowing that the rest of L2 is predominantly hidden below the surface of native virions (Buck CB, J Virol 2008). The authors should discuss it in their manuscript.

The above points has been included in the discussion as follows

Earlier reports suggest that the L2 protein is majorly buried or hidden under the surface of native and matured virions [85, 86]. The initial interactions between L1/L2 are hydrophobic with coverage of small stretches of amino acid sequences. It exhibits potential effects during in vitro assembly [87]. However, the structural relation of L2 minor capsid protein to L1 in the virion particles is not clearly known. In another study, Henio et al. [88] reported that the L1/L2 proteins of HPV have various antigenic epitope segments such as 32-81, 212-231, 272-291, and 347-381 amino acids, and could be accessible on the surface of L1/L2 virus-like particles..

3)      I am not fully convinced by the structural modelling performed by the authors. Indeed, many outliers are visible in the Ramachandran plot and the model should reach a score >95% for all residues present in Ramachandran favored regions. The errata plot also indicates some general problem with structural modeling with several residue over the rejection limit of 95%.It would also be useful for the authors to show the ERRATA on the full-length model and not only the first 140 amino acids.

The Ramachandran plot and ERRAT plot has been corrected and given in Figure 2

4)      Therefore, I invite the authors to perform a Homology modeling using another algorithm-based software. I would propose Modeler (B. Webb et al. Current Protocols in Bioinformatics 2016) or an ab initio prediction using Rosetta (Ovchinnikov S, Elife. 2015) or Robetta software (http://robetta.bakerlab.org/) for the two epitopes chosen to see how the model generated by these algorithms compare to the I-Tasser model.

The homology modeling for the constructed vaccine segments has been performed by using Robetta and the results has been compared with i-Tasser and shown in the Table

The Robetta beta server was used to predict the full chain protein structure. This server (http://robetta.bakerlab.org) gives automated tools for analysis and prediction of the protein structure. Robetta provides both ab initio and comparative models of protein domains. Comparative models are built from structures detected and aligned by HHSEARCH, SPARKS, and Raptor. Loop regions are assembled from fragments and optimized to fit the aligned template structures. De novo models are built using the Robetta de novo protocol. For structure prediction, the submitted query sequences were analyzed minutely into putative domains. For domain prediction, a hierarchical screening method called “Ginzu” was used [62].

Results: After modeling with Robetta, we obtained five different models for TLR5 and the chimeric vaccine candidate. Models 1 and 4 are de novo models while models 2, 3, and 5 are ab initio models. The Ginzu domain prediction confidence score for TLR5 is 0.9375 and for SGD58 it is 0.6502. All the models were selected for structural refinement analysis.

Minor points:

5)      There are some typos in the text that should be corrected.

The typographical errors has been corrected

6)      Figure 2 could be improved (colors)

The Figure 2 has been improved and provided in colors

7)      3- For Figure 3, it is very difficult to see anything. The authors should show the docking results from different angles. Front, top… and increase the resolution of Fig3b.

Figure 3. Ligplot prepared interacting residues in the TLR5-SGD58. (a, b, c, d) represents best structure of TLR5-SGD58 cluster. The color-coding represents the TLR5 in brown color, SGD58 in pink color. Dashed lines in green color denote hydrogen-bonding interactions.

Round  2

Reviewer 2 Report

In this revised manuscript, the authors have addressed satisfactorily all the points I raised.